# Generalization in Reinforcement Learning with Selective Noise Injection and Information Bottleneck

**Maximilian Igl** *
University of Oxford

**Kamil Ciosek**
Microsoft Research

**Yingzhen Li**
Microsoft Research

**Sebastian Tschiatschek**
Microsoft Research

**Cheng Zhang**
Microsoft Research

**Sam Devlin** †
Microsoft Research

**Katja Hofmann** †
Microsoft Research

## Abstract

The ability for policies to generalize to new environments is key to the broad application of RL agents. A promising approach to prevent an agent's policy from overfitting to a limited set of training environments is to apply regularization techniques originally developed for supervised learning. However, there are stark differences between supervised learning and RL. We discuss those differences and propose modifications to existing regularization techniques in order to better adapt them to RL. In particular, we focus on regularization techniques relying on the injection of noise into the learned function, a family that includes some of the most widely used approaches such as Dropout and Batch Normalization. To adapt them to RL, we propose *Selective Noise Injection* (SNI), which maintains the regularizing effect the injected noise has, while mitigating the adverse effects it has on the gradient quality. Furthermore, we demonstrate that the Information Bottleneck (IB) is a particularly well suited regularization technique for RL as it is effective in the low-data regime encountered early on in training RL agents. Combining the IB with SNI, we significantly outperform current state of the art results, including on the recently proposed generalization benchmark *Coinrun*.

## 1   Introduction

Deep Reinforcement Learning (RL) has been used to successfully train policies with impressive performance on a range of challenging tasks, including Atari [6, 16, 31], continuous control [35, 46] and tasks with long-ranged temporal dependencies [33]. In those settings, the challenge is to be able to successfully explore and learn policies complex enough to solve the training tasks. Consequently, the focus of these works was to improve the learning performance of agents in the training environment and less attention was being paid to generalization to testing environments.

However, being able to generalize is a key requirement for the broad application of autonomous agents. Spurred by several recent works showing that most RL agents overfit to the training environment [15, 58, 62, 65, 66], multiple benchmarks to evaluate the generalization capabilities of agents were proposed, typically by procedurally generating or modifying levels in video games [8, 11, 20, 21, 32, 63]. How to learn generalizable policies in these environments remains an open question, but early results have shown the use of regularization techniques (like weight decay, dropout and batch normalization) established in the supervised learning paradigm can also be useful for RL agents [11]. Our work builds on these results, but highlights two important differences between supervised learning and RL which need to be taken into account when regularizing agents.

First, because in RL the *training data depends on the model* and, consequently, the regularization method, stochastic regularization techniques like Dropout or BatchNorm can have adverse effects. For example, injecting stochasticity into the policy can lead to prematurely ending episodes, preventing the agent from observing future rewards. Furthermore, stochastic regularization can destabilize training through the learned *critic* and off-policy importance weights. To mitigate those adverse effects and effectively apply stochastic regularization techniques to RL, we propose Selective Noise Injection (SNI). It selectively applies stochasticity only when it serves regularization and otherwise computes the output of the regularized networks deterministically. We focus our evaluation on Dropout and the Variational Information Bottleneck (VIB), but the proposed method is applicable to most forms of stochastic regularization.

A second difference between RL and supervised learning is the *non-stationarity of the data-distribution* in RL. Despite many RL algorithms utilizing millions or even billions of observations, the diversity of states encountered early on in training can be small, making it difficult to learn general features. While it remains an open question as to why deep neural networks generalize despite being able to perfectly memorize the training data [5, 64], it has been shown that the optimal point on the worst-case generalization bound requires the model to rely on a more compressed set of features the fewer data-points we have [47, 54]. Therefore, to bias our agent towards more general features even early on in training, we adapt the Information Bottleneck (IB) principle to an actor-critic agent, which we call Information Bottleneck Actor Critic (IBAC). In contrast to other regularization techniques, IBAC directly incentivizes the compression of input features, resulting in features that are more robust under a shifting data-distribution and that enable better generalization to held-out test environments.

We evaluate our proposed techniques using Proximal Policy Optimization (PPO), an off-policy actor-critic algorithm, on two challenging generalization tasks, *Multiroom* [9] and *Coinrun* [11]. We show the benefits of both IBAC and SNI individually as well as in combination, with the resulting IBAC-SNI significantly outperforming the previous state of the art results.

## 2    Background

We consider having a distribution $q(m)$ of Markov decision processes (MDPs) $m \in \mathcal{M}$, with $m$ being a tuple $(\mathcal{S}_m, \mathcal{A}, T_m, R_m, p_m)$ consisting of state-space $\mathcal{S}_m$, action-space $\mathcal{A}$, transition distribution $T_m(s'|s, a)$, reward function $R_m(s, a)$ and initial state distribution $p_m(s_0)$ [38]. For training, we either assume unlimited access to $q(m)$ (like in section 5.2, *Multiroom*) or restrict ourselves to a fixed set of training environments $M_{\text{train}} = \{m_1, \ldots, m_n\}$, $m_i \sim q$ (like in section 5.3, *Coinrun*).

The goal of the learning process is to find a policy $\pi_\theta(a|s)$, parameterized by $\theta$, which maximizes the discounted expected reward: $J(\pi_\theta) = \mathbb{E}_{q, \pi, T_m, p_m} \left[ \sum_{t=0}^T \gamma^t R_m(s_t, a_t) \right]$. Although any RL method with an off-policy correction term could be used with our proposed method of SNI, PPO [46] has shown strong performance and enables direct comparison with prior work [11]. The actor-critic version of this algorithm collects trajectory data $\mathcal{D}_\tau$ using a rollout policy $\pi_\theta^r(a_t|s_t)$ and subsequently optimizes a surrogate loss:

$$L_{\text{PPO}} = -\mathbb{E}_{\mathcal{D}_\tau} \left[ \min(c_t(\theta) A_t, \text{clip}(c_t(\theta), 1 - \epsilon, 1 + \epsilon) A_t) \right] \tag{1}$$

with $c_t(\theta) = \frac{\pi_\theta(a_t|s_t)}{\pi_\theta^r(a_t|s_t)}$ for $K$ epochs. The advantage $A_t$ is computed as in A2C [30]. This is an efficient approximate trust region method [44], optimizing a pessimistic lower bound of the objective function on the collected data. It corresponds to estimating the gradient w.r.t the policy conservatively, since moving $\pi_\theta$ further away from $\pi_\theta^r$, such that $c_t(\theta)$ moves outside a chosen range $[1 - \epsilon, 1 + \epsilon]$, is only taken into account if it decreases performance. Similarly, the value function loss minimizes an upper bound on the squared error:

$$L_{\text{PPO}}^V = \mathbb{E}_{\mathcal{D}_\tau} \left[ \frac{1}{2} \max \left( (V_\theta - V^T)^2, (V^r + \text{clip}(V_\theta - V^r, 1 - \epsilon, 1 + \epsilon) - V_{\text{target}})^2 \right) \right] \tag{2}$$

with a bootstrapped value function target $V_{\text{target}}$ [30] and previous value function $V^r$. The overall minimization objective is then:

$$L_t(\theta) = L_{\text{PPO}} + \lambda_V L_{\text{PPO}}^V - \lambda_H H[\pi_\theta] \tag{3}$$

where $H[\cdot]$ denotes an entropy bonus to encourage exploration and prevent the policy to collapse prematurely. In the following, we discuss regularization techniques that can be used to mitigate overfitting to the states and MDPs so far seen during training.

## 2.1 Regularization Techniques in Supervised Learning

In supervised learning, classifiers are often regularized using a variety of techniques to prevent overfitting. Here, we briefly present several major approaches which we either utilize as baseline or extend to RL in section 4.

*Weight decay*, also called L2 regularization, reduces the magnitude of the weights $\theta$ by adding an additional loss term $\lambda_w \frac{1}{2}\|\theta\|_2^2$. With a gradient update of the form $\theta \leftarrow \theta - \alpha \nabla_\theta(L(\theta) + \frac{\lambda_w}{2}\|\theta\|_2^2)$, this decays the weights in addition to optimizing $L(\theta)$, i.e. we have $\theta \leftarrow (1 - \alpha\lambda_w)\theta - \alpha\nabla_\theta L(\theta)$.

*Data augmentation* refers to changing or distorting the available input data to improve generalization. In this work, we use a modified version of cutout [12], proposed by [11], in which a random number of rectangular areas in the input image is filled by random colors.

*Batch Normalization* [17, 18] normalizes activations of specified layers by estimating their mean and variance using the current mini-batch. Estimating the batch statistics introduces noise which has been shown to help improve generalization [28] in supervised learning.

Another widely used regularization technique for deep neural networks is *Dropout* [48]. Here, during training, individual activations are randomly zeroed out with a fixed probability $p_d$. This serves to prevent co-adaptation of neurons and can be applied to any layer inside the network. One common choice, which we are following in our architecture, is to apply it to the last hidden layer.

Lastly, we will briefly describe the Variational Information Bottleneck (VIB) [2], a deep variational approximation to the Information Bottleneck (IB) [53]. While not typically used for regularization in deep supervised learning, we demonstrate in section 5 that our adaptation IBAC shows strong performance in RL. Given a data distribution $p(X, Y)$, the learned model $p_\theta(y|x)$ is regularized by inserting a stochastic latent variable $Z$ and minimizing the mutual information between the input $X$ and $Z$, $I(X, Z)$, while maximizing the predictive power of the latent variable, i.e. $I(Z, Y)$. The VIB objective function is:

$$L_{\text{VIB}} = \mathbb{E}_{p(x,y), p_\theta(z|x)}\big[ -\log q_\theta(y|z) + \beta D_{KL}[p_\theta(z|x)\|q(z)]\big] \tag{4}$$

where $p_\theta(z|x)$ is the encoder, $q_\theta(y|z)$ the decoder, $q(z)$ the approximated latent marginal often fixed to a normal distribution $\mathcal{N}(0, I)$ and $\beta$ is a hyperparameter. For a normal distributed $p_\theta(z|x)$, eq. (4) can be optimized by gradient decent using the reparameterization trick [25].

## 3 The Problem of Using Stochastic Regularization in RL

We now take a closer look at a prototypical objective for training actor-critic methods and highlight important differences to supervised learning. Based on those observations, we propose an explanation for the finding that some stochastic optimization methods are less effective [11] or can even be detrimental to performance when combined with other regularization techniques (see appendix D).

In supervised learning, the optimization objective takes a form similar to $\max_\theta \mathbb{E}_\mathcal{D}\big[\log p_\theta(y|x)\big]$, where we highlight the model $p_\theta(y|x)$ to be updated in blue, $\mathcal{D}$ is the available data and $\theta$ the parameters to be learned. On the other hand, in RL the objective for the actor is to maximize $J(\pi_\theta) = \mathbb{E}_{\pi_\theta(a|s)}\big[\sum_t \gamma^t R_m(s_t, a_t)\big]$, where, for convencience, we drop $q$, $T_m$ and $p_m$ from the notation of the expectation. Because now the learned distribution, $\pi_\theta(a|s)$, is part of data-generation, computing the gradients, as done in policy gradient methods, requires the log-derivative trick. For the class of deep off-policy actor-critic methods we are experimentally evaluating in this paper, one also typically uses the policy gradient theorem [52] and an estimated *critic* $V_\theta(s)$ as baseline and for bootstrapping to reduce the gradient variance. Consequently, the gradient estimation becomes:

$$\nabla_\theta J(\pi_\theta) = \mathbb{E}_{\pi_\theta^r(a_t|s_t)}\left[\sum_t^T \frac{\pi_\theta(a_t|s_t)}{\pi_\theta^r(a_t|s_t)}\nabla_\theta \log \pi_\theta(a_t|s_t)(r_t + \gamma V_\theta(s_{t+1}) - V_\theta(s_t))\right] \tag{5}$$

where we utilize a *rollout policy* $\pi_\theta^r$ to collect trajectories. It can deviate from $\pi_\theta$ but should be similar to keep the off-policy correction term $\pi_\theta/\pi_\theta^r$ low variance. In eq. (5), only the term $\pi_\theta(a_t|s_t)$ is being updated and we highlight in orange all the additional influences of the learned policy and critic on the gradient.

Denoting by the superscript $\perp$ that $V_\theta^\perp$ is assumed constant, we can write the optimization objective for the critic as

$$L_{\text{AC}}^V = \min_\theta \mathbb{E}_{\pi_\theta^r(a_t|s_t)} \left[ \left( \gamma V_\theta^\perp(s_{t+1}) + r_t - V_\theta(s_t) \right)^2 \right] \tag{6}$$

From eqs. (5) and (6) we can see that the injection of noise into the computation of $\pi_\theta^r$ and $V_\theta$ can degrade performance in several ways: i) During rollouts using the rollout policy $\pi_\theta^r$, it can lead to undesirable actions, potentially ending episodes prematurely, and thereby deteriorating the quality of the observed data; ii) It leads to a higher variance of the off-policy correction term $\pi_\theta/\pi_\theta^r$ because the injected noise can be different for $\pi_\theta$ and $\pi_\theta^r$, increasing gradient variance; iii) It increases variance in the gradient updates of both the policy and the critic through variance in the computation of $V_\theta$.

## 4 Method

To utilize the strength of noise-injecting regularization techniques in RL, we introduce Selective Noise Injection (SNI) in the following section. Its goal is to allow us to make use of such techniques while mitigating the adverse effects the added stochasticity can have on the RL gradient computation. Then, in section 4.2, we propose Information Bottleneck Actor Critic (IBAC) as a new regularization method and detail how SNI applies to IBAC, resulting in our state-of-the art method IBAC-SNI.

### 4.1 Selective Noise Injection

We have identified three sources of negative effects due to noise which we need to mitigate: In the rollout policy $\pi_\theta^r$, in the critic $V_\theta$ and in the off-policy correction term $\pi_\theta/\pi_\theta^r$. We first introduce a short notation for eq. (5) as $\nabla_\theta J(\pi_\theta) = \mathcal{G}_{\text{AC}}(\pi_\theta^r, \pi_\theta, V_\theta)$.

To apply SNI to a regularization technique relying on noise-injection, we need to be able to *temporarily suspend* the noise and compute the output of the model deterministically. This is possible for most techniques[3]. For example, in Dropout, we can freeze one particular dropout mask, in VIB we can pass in the mode instead of sampling from the posterior distribution and in Batch Normalization we can either utilize the moving average instead of the batch statistics or freeze and re-use one statistic multiple times. Formally, we denote by $\bar{\pi}_\theta$ the version of a component $\pi_\theta$, with the injected regularization noise suspended. Note that this does not mean that $\bar{\pi}_\theta$ is deterministic, for example when the network approximates the parameters of a distribution.

Then, for SNI we modify the policy gradient loss as follows: i) We use $\bar{V}_\theta$ as critic instead of $V_\theta$ in both eqs. (5) and (6), eliminating unnecessary noise through the critic; ii) We use $\bar{\pi}^r$ as rollout policy instead of $\pi^r$. For some regularization techniques this will reduce the probability of undesirable actions; iii) We compute the policy gradient as a *mixture* between gradients for $\pi_\theta$ and $\bar{\pi}_\theta$ as follows:

$$\mathcal{G}_{\text{AC}}^{\text{SNI}}(\pi_\theta^r, \pi_\theta, V_\theta) = \lambda \mathcal{G}_{\text{AC}}(\bar{\pi}_\theta^r, \bar{\pi}_\theta, \bar{V}_\theta) + (1-\lambda)\mathcal{G}_{\text{AC}}(\bar{\pi}_\theta^r, \pi_\theta, \bar{V}_\theta) \tag{7}$$

The first term guarantees a lower variance of the off-policy importance weight, which is especially important early on in training when the network has not yet learned to compensate for the injected noise. The second term uses the noise-injected policy for updates, thereby taking advantage of its regularizing effects while still reducing unnecessary variance through the use of $\bar{\pi}^r$ and $\bar{V}_\theta$. Note that sharing the rollout policy $\bar{\pi}^r$ between both terms allows us to use the same collected data. Furthermore most computations are shared between both terms or can be parallelized.

### 4.2 Information Bottleneck Actor Critic

Early on in training an RL agent, we are often faced with little variation in the training data. Observed states are distributed only around the initial states $s_0$, making spurious correlations in the low amount of data more likely. Furthermore, because neither the policy nor the critic have sufficiently converged yet, we have a high variance in the target values of our loss function.

This combination makes it harder and less likely for the network to learn desirable features that are robust under a shifting data-distribution during training and generalize well to held-out test MDPs.

To counteract this reduced signal-to-noise ratio, our goal is to explicitly bias the learning towards finding more *compressed* features which are shown to have a tighter worst-case generalization bound [54]. While a higher compression does not guarantee robustness under a *shifting* data-distribution, we believe this to be a reasonable assumption in the majority of MDPs, for example because they rely on a consistent underlying transition mechanism like physical laws.

To incentivize more compressed features, we use an approach similar to the VIB [2], which minimizes the mutual information $I(S, Z)$ between the state $S$ and its latent representation $Z$ while maximizing $I(Z, A)$, the predictive power of $Z$ on actions $A$. To do so, we re-interpret the policy gradient update as maximization of the log-marginal likelihood of $\pi_\theta(a|s)$ under the data distribution $p(s, a) := \frac{\rho^\pi(s)\pi_\theta(a|s)A^\pi(s,a)}{\mathcal{Z}}$ with discounted state distribution $\rho^\pi(s)$, advantage function $A^\pi(s, a)$ and normalization constant $\mathcal{Z}$. Taking the semi-gradient of this objective, i.e. assuming $p(s, a)$ to be fixed, recovers the policy gradient:

$$\nabla_\theta \mathcal{Z} \, \mathbb{E}_{p(s,a)}[\log \pi_\theta(a|s)] = \int \rho^\pi(s)\pi_\theta(a|s)\nabla_\theta \log \pi_\theta(a|s)A^\pi(s, a) \, \mathrm{d}s \, \mathrm{d}a. \tag{8}$$

Now, following the same steps as [2], we introduce a stochastic latent variable $z$ and minimize $\beta I(S, Z)$ while maximizing $I(Z, A)$ under $p(s, a)$, resulting in the new objective:

$$L_{\text{IB}} = \mathbb{E}_{p(s,a),p_\theta(z|s)}\big[-\log q_\theta(a|z) + \beta D_{KL}[p_\theta(z|s)\|q(z)]\big] \tag{9}$$

We take the gradient and use the reparameterization trick [25] to write the encoder $p_\theta(z|s)$ as deterministic function $f_\theta(s, \epsilon)$ with $\epsilon \sim p(\epsilon)$:

$$\begin{aligned}\nabla_\theta L_{\text{IB}} &= -\mathbb{E}_{\rho^\pi(s)\pi_\theta(a|s)p(\epsilon)}\big[\nabla_\theta \log q_\theta(a|f_\theta(s, \epsilon))A^\pi(s, a)\big] + \nabla_\theta \beta D_{KL}[p_\theta(z|s)\|q(z)] \\ &= \nabla_\theta(L_{\text{AC}}^{\text{IB}} + \beta L^{\text{KL}}),\end{aligned} \tag{10}$$

resulting in a modified policy gradient objective and an additional regularization term $L^{\text{KL}}$.

Policy gradient algorithms heuristically add an entropy bonus $H[\pi_\theta(a|s)]$ to prevent the policy distribution from collapsing. However, this term also influences the distributions over $z$. In practice, we are only interested in preventing $q_\theta(a|z)$ (not $\pi_\theta(a|s) = \mathbb{E}_z[q_\theta(a|z)]$) from collapsing because our rollout policy $\bar{\pi}_\theta$ will not rely on stochasticity in $z$. Additionally, $p_\theta(z|s)$ is already entropy-regularized by the IB loss term[4]. Consequently, we adapt the heuristic entropy bonus to

$$H^{\text{IB}}[\pi_\theta(a|s)] := \int p_\theta(s, z)H[q_\theta(a|z)] \, \mathrm{d}s \, \mathrm{d}z, \tag{11}$$

resulting in the overall loss function of the proposed Information Bottleneck Actor Critic (IBAC)

$$L_t^{\text{IBAC}}(\theta) = L_{\text{AC}}^{\text{IB}} + \lambda_V L_{\text{AC}}^V - \lambda_H H^{\text{IB}}[\pi_\theta] + \beta L^{\text{KL}} \tag{12}$$

with the hyperparameters $\lambda_V$, $\lambda_H$ and $\beta$ balancing the loss terms.

While IBAC incentivizes more compressed features, it also introduces stochasticity. Consequently, combining it with SNI improves performance, as we demonstrate in sections 5.2 and 5.3. To compute the noise-suspended policy $\bar{\pi}_\theta$ and critic $\bar{V}_\theta$, we use the mode $z = \mu_\theta(s)$ as input to $q_\theta(a|z)$ and $V_\theta(z)$, where $\mu_\theta(s)$ is the mode of $p_\theta(z|s)$ and $V_\theta(z)$ now conditions on $z$ instead of $s$, also using the compressed features. Note that for SNI with $\lambda = 1$, i.e. with only the term $\mathcal{G}_{\text{AC}}(\bar{\pi}_\theta^r, \bar{\pi}_\theta, \bar{V}_\theta)$, this effectively recovers a L2 penalty on the activations since the variance of $z$ will then always be ignored and the KL-divergence between two Gaussians minimizes the squared difference of their means.

## 5 Experiments

In the following, we present a series of experiments to show that the IB finds more general features in the low-data regime and that this translates to improved generalization in RL for IBAC agents, especially when combined with SNI. We evaluate our proposed regularization techniques on two environments, one grid-world with challenging generalization requirements [9] in which most previous approaches are unable to find the solution and on the recently proposed *Coinrun* benchmark [11]. We show that IBAC-SNI outperforms previous state of the art on both environments by a large margin. Details about the used hyperparameters and network architectures can be found in the Appendix, code to reproduce the results can be found at https://github.com/microsoft/IBAC-SNI/.

## 5.1 Learning Features in the Low-Data Regime

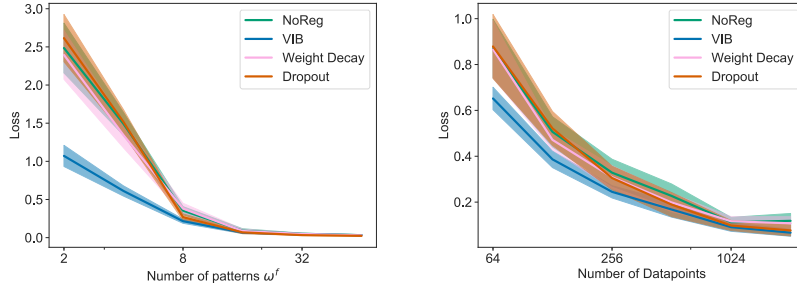

Figure 1: We show the loss on the test-data (lower is better). *Left:* Higher $\omega^f$ result in a larger difference in generality between features $f^c$ and $g^c$, making it easier to fit to the more general $g^c$. *Right:* Learning $g^c$ with fewer datapoints is more challenging, but needed early in training RL agents.

First we start in the supervised setting and show on a synthetic dataset that the VIB is particularly strong at finding more general features in the low-data regime and in the presence of multiple signals with varying degrees of generality. Our motivation is that the low-data regime is commonly encountered in RL early on in training and many environments allow the agent to base its decision on a variety of features in the state, of which we would like to find the most general ones.

We generate the training dataset $\mathcal{D}_{\text{train}} = \{(c_i, x_i)\}_{i=1}^N$ with observations $x_i \in \mathbb{R}^{d_x}$ and classes $c_i \in \{1, \ldots, n_c\}$. Each data point $i$ is generated by first drawing the class $c_i \sim Cat(n_c)$ from a uniform categorical distribution and generating the vector $x_i$ by embedding the information about $c_i$ in *two* different ways $g^c$ and $f^c$ (see appendix B for details). Importantly, only $g^c$ is shared between the training and test set. This allows us to measure the model's relative reliance on $g^c$ and $f^c$ by measuring the test performance (all models perfectly fit the training data). We allow $f^c$ to encode the information about $c_i$ in $\omega^f$ different ways. Consequently, the higher $\omega^f$, the less general $f^c$ is.

In fig. 1 we measure how the test performance of fully trained classification models varies for different regularization techniques when we i) vary the generality of $f^c$ and ii) vary the number of data-points in the training set. We find that most techniques perform comparably with the exception of the VIB which is able to find more general features both in the low-data regime and in the presence of multiple features with only small differences in generality. In the next section, we show that this translates to faster training and performance gains in RL for our proposed algorithm IBAC.

## 5.2 Multiroom

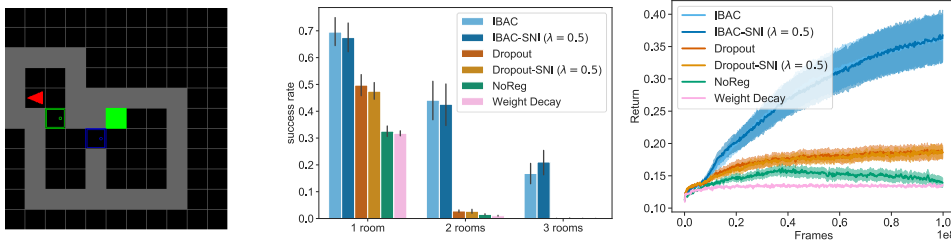

Figure 2: *Left:* Typical layout of the environment. The red triangle denotes the agent and its direction, the green full square is the goal, colored boxes are doors and grey squares are walls. *Middle:* Probability of finding the goal depending on level size for models trained on all levels. Shown are mean and standard error across 30 different seeds. *Right:* Mean and standard error over of the return of the same models averaged across all room sizes.

In this section, we show how IBAC can help learning in RL tasks which require generalization. For this task, we do not distinguish between training and testing, but for each episode, we draw $m$ randomly from the full distribution over MDPs $q(m)$. As the number of MDPs is very large, learning can only be successful if the agent learns general features that are transferrable between episodes.

This experiment is based on [9]. The aim of the agent is to traverse a sequence of rooms to reach the goal (green square in fig. 2) as quickly as possible. It takes discrete actions to rotate $90°$ in either direction, move forward and toggle doors to be open or closed. The observation received by the agent includes the full grid, one pixel per square, with object type and object status (like direction) encoded in the 3 color channels. Crucially, for each episode, the layout is generated randomly by placing a random number of rooms $n_r \in \{1, 2, 3\}$ in a sequence connected by one door each.

The results in fig. 2 show that IBAC agents are much better at successfully learning to solve this task, especially for layouts with more rooms. While all other fully trained agents can solve less than 3% of the layouts with two rooms and none of the ones with three, IBAC-SNI still succeeds in an impressive 43% and 21% of those layouts. The difficulty of this seemingly simple task arises from its generalization requirements: Since the layout is randomly generated in each episode, each state is observed very rarely, especially for multi-room layouts, requiring generalization to allow learning. While in the 1 room layout the reduced policy stochasticity of the SNI agent slightly reduces performance, it improves performance for more complex layouts in which higher noise becomes detrimental. In the next section we will see that this also holds for the much more complex *Coinrun* environment in which SNI significantly improves the IBAC performance.

## 5.3   Coinrun

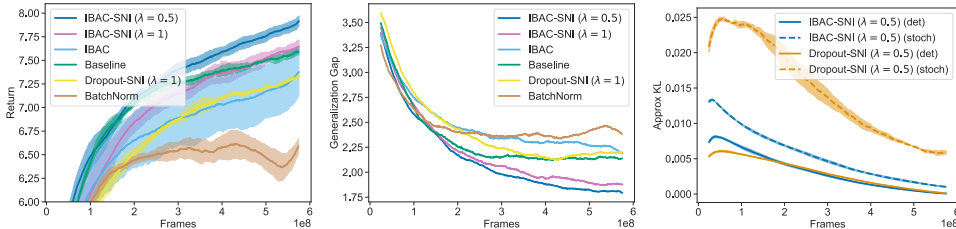

Figure 3: *Left:* Performance of various agents on the test environments. We note that 'BatchNorm' corresponds to the best performing agent in [11]. Furthermore, 'Dropout-SNI ($\lambda = 1$)' is similar to the Dropout implementation used in [11] but was previously not evaluated with weight decay and data augmentation. *Middle:* Difference between test performance and train performance (see fig. 7). Without standard deviation for readability. *Right:* Averaged approximate KL-Divergence between rollout policy and updated policy, used as proxy for the variance of the importance weight. Mean and standard deviation are across three random seeds.

On the previous environment, we were able to show that IBAC and SNI help agents to find more general features and to do so faster. Next, we show that this can lead to a higher final performance on previously unseen test environments. We evaluate our proposed regularization techniques on *Coinrun* [11], a recently proposed generalization benchmark with high-dimensional observations and a large variety in levels. Several regularization techniques were previously evaluated there, making it an ideal evaluation environment for IBAC and SNI. We follow the setting proposed in [11], using the same 500 levels for training and evaluate on randomly drawn, new levels of only the highest difficulty.

As [11] have shown, combining multiple regularization techniques can improve performance, with their best- performing agent utilizing data augmentation, weight decay and batch normalization. As our goal is to push the state of the art on this environment and to accurately compare against their results, fig. 3 uses weight decay and data-augmentation on all experiments. Consequently, 'Baseline' in fig. 3 refers to *only* using weight decay and data-augmentation whereas the other experiments use Dropout, Batch Normalization or IBAC *in addition* to weight decay and data-augmentation. Results without those baseline techniques can be found in appendix D.

First, we find that almost all previously proposed regularization techniques *decrease* performance compared to the baseline, see fig. 3 (left), with batch normalization performing worst, possibly due to its unusual interaction with weight decay [56]. Note that this combination with batch normalization was the highest performing agent in [11]. We conjecture that regularization techniques relying on stochasticity can introduce additional instability into the training update, possibly deteriorating performance, especially if their regularizing effect is not sufficiently different from what weight decay and data-augmentation already achieve. This result applies to both batch normalization and Dropout,

with and without SNI, although SNI mitigates the adverse effects. Consequently, we can already improve on the state of the art by only relying on those two non-stochastic techniques. Furthermore, we find that IBAC in combination with SNI is able to significantly outperform our new state of the art baseline. We also find that for IBAC, $\lambda = 0.5$ achieves better performance than $\lambda = 1$, justifying using both terms in eq. (7).

As a proxy for the variance of the off-policy correction term $\pi_\theta^r / \pi_\theta$, we show in fig. 3 (right) the estimated, averaged KL-divergence between the rollout policy and the update policy for both terms, $\mathcal{G}_{AC}(\bar{\pi}_\theta^r, \bar{\pi}_\theta, \bar{V}_\theta)$, denoted by '(det)' and $\mathcal{G}_{AC}(\bar{\pi}_\theta^r, \pi_\theta, \bar{V}_\theta)$, denoted by '(stoch)'. Because PPO uses data-points multiple times it is non-zero even for the deterministic term. First, we can see that using the deterministic version reduces the KL-Divergence, explaining the positive influence of $\mathcal{G}_{AC}(\bar{\pi}_\theta^r, \bar{\pi}_\theta, \bar{V}_\theta)$. Second, we see that the KL-Divergence of the stochastic part is much higher for Dropout than for IBAC, offering an explanation of why for Dropout relying on purely the deterministic part ($\lambda = 1$) outperforms an equal mixing $\lambda = 0.5$ (see fig. 6).

## 6   Related Work

Generalization in RL can take a variety of forms, each necessitating different types of regularization. To position this work, we distinguished two types that, whilst not mutually exclusive, we believe to be conceptually distinct and found useful to isolate when studying approaches to improve generalization.

The first type, *robustness to uncertainty* refers to settings in which the unobserved MDP $m$ influences the transition dynamics or reward structure. Consequently the current state $s$ might not contain enough information to act optimally in the current MDP and we need to find the action which is optimal under the uncertainty about $m$. This setting often arises in robotics and control where exact physical characteristics are unknown and domain shifts can occur [27]. Consequently, *domain randomization*, the injection of randomness into the environment, is often purposefully applied during training to allow for sim-to-real transfer [26, 55]. Noise can be injected into the states of the environment [50] or the parameters of the transition distribution like friction coefficients or mass values [3, 34, 60]. The noise injected into the dynamics can also be manipulated adversarially [29, 36, 39]. As the goal is to prevent overfitting to specific MDPs, it also has been found that using smaller [40] or simpler [66] networks can help. We can also aim to learn an adaptive policy by treating the environment as partially observable Markov decision process (POMDP) [35, 60] (similar to viewing the learning problem in the framework of Bayesian RL [37]) or as a meta-learning problem [1, 10, 13, 43, 51, 57].

On the other hand, we distinguish *feature robustness*, which applies to environments with high-dimensional observations (like images) in which generalization to previously unseen states can be improved by learning to extract better features, as the focus for this paper. Recently, a range of benchmarks, typically utilizing procedurally generated levels, have been proposed to evaluate this type of generalization [4, 11, 19, 20, 21, 22, 32, 59, 63].

Improving generalization in those settings can rely on generating more diverse observation data [11, 42, 55], or strong, often relational, inductive biases applied to the architecture [23, 49, 61]. Contrary to the results in continuous control domains, here deeper networks have been found to be more successful [7, 11]. Furthermore, this setting is more similar to that of supervised learning, so established regularization techniques like weight decay, dropout or batch-normalization have also successfully been applied, especially in settings with a limited number of training environments [11]. This is the work most closely related to ours. We build on those results and improve upon them by taking into account the specific ways in which RL is *different* from the supervised setting. They also do not consider the VIB as a regularization technique.

Combining RL and VIB has been recently explored for learning goal-conditioned policies [14] and meta-RL [41]. Both of these previous works [14, 41] also differ from the IBAC architecture we propose by conditioning action selection on both the encoded and raw state observation. These studies complement the contribution made here by providing evidence that the VIB can be used with a wider range of RL algorithms including demonstrated benefits when used with Soft Actor-Critic for continuous control in MuJoCo [41] and on-policy A2C in MiniGrid and MiniPacMan [14].

## 7  Conclusion

In this work we highlight two important differences between supervised learning and RL: First, the training data is generated using the learned model. Consequently, using stochastic regularization methods can induce adverse effects and reduce the quality of the data. We conjecture that this explains the observed lower performance of Batch Normalization and Dropout. Second, in RL, we often encounter a noisy, low-data regime early on in training, complicating the extraction of general features.

We argue that these differences should inform the choice of regularization techniques used in RL. To mitigate the adverse effects of stochastic regularization, we propose Selective Noise Injection (SNI) which only selectively injects noise into the model, preventing reduced data quality and higher gradient variance through a noisy critic. On the other hand, to learn more compressed and general features in the noisy low-data regime, we propose Information Bottleneck Actor Critic (IBAC), which utilizes an variational information bottleneck as part of the agent.

We experimentally demonstrate that the VIB is able to extract better features in the low-data regime and that this translates to better generalization of IBAC in RL. Furthermore, on complex environments, SNI is key to good performance, allowing the combined algorithm, IBAC-SNI, to achieve state of the art on challenging generalization benchmarks. We believe the results presented here can inform a range of future works, both to improve existing algorithms and to find new regularization techniques adapted to RL.

## Acknowledgments

We would like to thank Shimon Whiteson for his helpful feedback, Sebastian Lee, Luke Harris, Hiske Overweg and Patrick Fernandes for help with experimental evaluations and Adrian O'Grady, Jaroslaw Rzepecki and Andre Kramer for help with the computing infrastructure. M. Igl is supported by the UK EPSRC CDT on Autonomous Intelligent Machines and Systems.

## Footnotes

*Work performed during an internship at Microsoft Research Cambridge

†Co-Senior Authors

[3]In this work, we will focus on VIB and Dropout as those show the most promising results without SNI (see section 5) and will leave its application to other regularization techniques for future work.

[4]We have $D_{\text{KL}}[p_\theta(z|s)\|r(z)] = \mathbb{E}_{p_\theta(z|s)}[\log p_\theta(z|s) - \log r(z)] = -H[p_\theta(z|s)] - \mathbb{E}_{p_\theta(z|s)}[\log r(z)]$

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
