[Supplementary Material · IBAC-SNI_CRC_appendix.pdf]

## A  Dropout with SNI

In order to apply SNI to Dropout, we need to decide how to 'suspend' the noise to compute $\bar{\pi}_\theta$. While one could apply no dropout mask and scale the activations accordingly, we empirically found it to be better to instead sample one dropout mask and keep it fixed for all gradient updates using the thus collected data. This follows the implementation used in [11].

## B  Supervised Classification Task

**Network architecture and hyperparameters**  The network consist of a 1D-convolutional layer with 10 filters and a kernel size of 11 followed by two hidden, fully connected layers of size 1024 and 256 and the last layer which outputs $n_c$ logits. When the VIB or Dropout are used, they are applied to the last hidden layer. We use a learning rate of $1e-4$. The relative weight for weight decay was $\lambda_w = 1e-3$, which performed best out of $\{1e-2, 1e-3, 1e-4, 1e-5\}$. For the VIB we used $\beta = 1e-3$, which performed best out of $\{1e-2, 1e-3, 1e-4, 1e-5\}$. Lastly, For dropout we tested the dropout rates $p_d \in \{0.1, 0.2, 0.5\}$, out of which $0.2$ performed best. Our results were stable across a range of hyperparameters, see fig. 5.

Figure 4: Generation of the input data $x$: We embed the information about $c$ twice, once through $f^c$ and once through $g^c$. See text for details.

**Data generation**  For each data-point, after drawing the class label $c_i$, we want to encode the information about $c_i$ in two ways, using the encoding functions $f^c$ and $g^c$ which use one of $\omega^f$ and $\omega^g$ different patterns to encode the information. The larger $\omega$ is, the less *general* the encoding is as it applies to fewer data-points. Note that there are $\omega$ different patterns *per class*.

We generate the patterns by first generating a set of random functions $\{f_j^c\}_{j=1}^{\omega^f}$ and $\{g_j^c\}_{j=1}^{\omega^g}$ by randomly drawing Fourier coefficients from $[0, 1]$. Those functions are converted into vectors by evaluating them at $d_x$ sorted points randomly drawn from $[0, 1]$. The resulting pattern-vectors for $f^c$ will have a dimension of $d_x$, whereas the ones for $g^c$ will be smaller, $d_g < d_x$.

To encode the information about $c_i$ we first choose one pattern from $\{f_j^c\}_j$ (slightly overloading notation between functions and pattern-vectors) and add some noise:

$$x_i' = f_j^c + \epsilon_i \quad \text{where} \quad \epsilon \sim \mathcal{N}(0, \sigma_\epsilon) \quad \text{and} \quad j \sim Cat(\omega^f) \tag{13}$$

Next, to also encode the information about $c_i$ using $g^c$, we choose one of the $\omega^g$ patterns $\{g_j^c\}_j$ and *replace a part of the vector $x_i'$*, which is possible because the $g^c$ patterns are shorter: $d_g < d_x$. The location of replacement is randomly drawn for each data-point, but restricted to a a set of $n_g$ possible locations which are also random, but kept fixed for the experiment and the same between training and testing set. The process is pictured in fig. 4.

By changing the number of possible locations $n_g$ and the strength of the noise added to $f^c$, $\sigma_\epsilon$, we can tune the relative difficulty of learning to recognize patterns $g^c$ and $f^c$, allowing us to find a regime where both *can* be found. Within this regime, our qualitative results were stable. We use $n_g = 3$ and $\sigma_\epsilon = 1$. Furthermore, we have for the dimension of of the observations $d_x = 100$, and for the size of the patterns $g^c$ we have $d_g = 20$. We use $n_c = 5$ different classes.

Figure 5: Loss function (error) on test set. Same results as in main text, but for multiple hyperparameters. The qualitative results are stable under a wide range of hyperparameters.

## C    Multiroom

The observation space measures $11 \times 11 \times 3$ where the 3 channels are used to encode object type and object features like orientation or 'open/closed' and 'color' for doors on each of the $11 \times 11$ spatial locations (see fig. 2 for a typical layout for $n_r = 3$).

The agent uses a 3-layer CNN with $16, 32$ and $32$ filters respectively. All layers use a kernel of size 2. After the CNN, it uses one hidden layer of size 64 to which IBAC or Dropout are applied if they are used. Dropout uses $p_d = 0.2$ and was tested for $\{0.1, 0.2, 0.5\}$. Both weight decay and IBAC were tried with a weighting factor of $\{1e-3, 1e-4, 1e-5, 1e-6\}$, with $1e-4$ performing best for weight decay and $1e-6$ performing best for IBAC. The output of the hidden layer is fed into a value function head and the policy head.

We use a discount factor $\gamma = 0.99$, a learning rate of $7e-4$, generalized value estimation with $\lambda_{\text{GAE}} = 0.95$ [45], an entropy coefficient of $\lambda_H = 0.01$, value loss coefficient $\lambda_V = 0.5$, gradient clipping at $0.5$ [45], and PPO with the Adam optimizer [24].

## D    Coinrun

**Architecture and Hyperparameters**    We use the same architecture ('Impala') and default policy gradient hyperparameters as well as the codebase (https://github.com/openai/coinrun) from the authors of [11] to ensure staying as closely as possible to their proposed benchmark.

Dropout and IBAC where applied to the last hidden layer and both, as well as weight decay, were tried with the same set of hyperparameters as in Multiroom. The best performance was achieved with $p_d = 0.2$ for Dropout and $1e-4$ for IBAC and weight decay. Batch normalization was applied between the layers of the convolutional part of the network. Note that the original architecture in [11] uses Dropout also on earlier layers, however, we achieve higher performance with our implementation.

In fig. 6 (left) we show results for Dropout with and without SNI and for $\lambda = 1$ and $\lambda = 0.5$. We find that $\lambda = 1$ learns fastest, possible due to the high importance weight variance in the stochastic

term in SNI for $\lambda < 1$ (see fig. 3 (right)). However, all Dropout implementations converge to roughly the same value, significantly below the 'baseline' agent, indicating that Dropout is not suitable for combination with weight decay and data augmentation.

In fig. 6 (right) we show the test performance for IBAC and Dropout with and without SNI, *without* using weight decay and data-augmentation. Again, we can see that SNI helps the performance. Interestingly, we can see that IBAC does not prevent overfitting by itself (one can see the performance decreasing for longer training) but does lead to faster learning. Our conjecture is that it finds more general features early on in training, but ultimately overfits to the test-set of environments without additional regularization. This further indicates that it's regularization is different to techniques such as weight decay, explaining why their combination synergizes well.

In fig. 7 we show the training set performance of our experiments.

Figure 6: *Left:* Comparison for different implementations of Dropout on the test environments. *Right:* Comparison of IBAC and Dropout, with and without SNI, *without weight decay and data augmentation*.

Figure 7: Training Performance with weight decay and data augmentation (left) and without (right)