[Reviews · NeurIPS 2019]

Reviewer 1



This work builds on the previous work about generalization in RL ([10] in the paper references) by (re-)investigating the classical stochastic regularization approaches in this context. It completes and updates the claims made in [10] by focusing of similar performance based experiments. Clarity: The method is clearly described in the paper. Significance: The question of generalization in RL is of great interest to the field. Main comments: - The paper motivates well the problems one faces when is comes to regularization in RL. The idea of being “selective” when injecting a regularizing noise makes a lot of convincing. However, the proposed method is more about an ad-hoc mixture of training modes, with noise injection and without, which makes the notion of selectivity here a bit loose. - Related to the previous point, since the mixture can be done in several ways, a comparison with other ways in order to justify this choice of the mixture can greatly enhance the paper quality. For example, one can imagine adapting the dropout probability or scheduling it in a curriculum learning fashion. How does your approach relate/compare to this ? - In the multiroom environment, it is confusing that your method has the highest variance while the method was presented as to be favoring variance reduction. Also, IBAC seems to have a higher trend than IBAC-SNI above 0.4 x 10^8 frames (IBAC has not stabilized like IBAC-SNI), which makes the fact that IBAC outperforms IBAC-SNI very likely when trained for longer than 10^8 frames. - It is not clear from eq. 7 how SNI circumvents some of the issues listed as motivation for it. For example, how does the roll-out avoid “undesirable actions (potentially ending episodes prematurely) ? - From the experiments point of view, a deeper analysis could be expected to relate to the issues listed as being specific to regularization in RL. The variance of the terms described in section 3, the nature of the trajectories ... etc ? In other words, it is not clear how SNI deals with the stated issues in section 3. Minor comments: - Should the normalization constant Z disappear in the right hand side of eq. 8 ? Also does this equation correspond to the semi-gradient or the exact gradient ? - Was L^V_{AC} defined in the paper ? and V_\theta(z) ? (I might have missed them) - In figure 3, can we assume that the curves with no std have very small variance ? Does the averaged curves over more runs for those one change the observations ? - When you say “outperforms an equal mixing = 0.5 early on in training ”, to which curve are you relating here ? Is it Dropout with \lambda=0.5 ? ------------------------------------------------------------------------ I have read the author response which has fairly addressed most of my concerns so i decided to increase my score. I would expect the authors to introduce more convincing plots of the claimed performances (like the one in the rebuttal) and a clearer justification of their choice of this type of mixture, in a future version of the paper.

Reviewer 2



The main contribution of the paper is to introduce a novel usage of the Information Bottleneck technique for off-policy actor-critic algorithms involving importance sampling and function approximation based critics, which results in state of the art results on the CoinRun challenge domain. An insightful discussion on some of the challenges when applying techniques that improve the generalization of supervised learning techniques to some RL settings. In general I think the paper is well motivated and supported sufficiently with experimental evidence. I have a couple of presentation concerns but generally think the paper is acceptable. Presentation: In general the paper is clearly written, however I do think there is a fundamental misunderstanding in the presentation, and think the authors would do well to avoid statements like "in RL the loss function depends on the learned parameters" or phrases like "the RL loss". In IID supervised learning, while one typically talks about having the same loss function (such as square loss) for all examples, with a relatively simple form, the loss function itself is a function of the parameters, so what is said isn't strictly correct. It's overly simplistic to view every single RL algorithm or technique as minimizing some loss function. Some algorithms are expressed in the form of weight update mechanisms, and rely on being contraction mappings or what not to converge, with function approximation being applied heuristically to scale to larger problems. Policy gradient techniques estimate gradient of the expected future return under some parametrized policy, it's not a loss function per-se, it's a goal you can estimate a gradient with respect to and make local improvements. Perhaps I am being overly pedantic, but I think the presentation would be more clear if you explicitly state the setting you are interested in (the empirically successful class of off-policy actor-critic algorithms involving importance sampling and function approximation based critics) and limit your broader statements and intuition to this setting. Too be clear, I have no problem with the motivation or insights behind the algorithmic changes, I just think the broader discussion is a bit sloppy. More minor points: - In line 66, the expectation should be over q not m. - Use q(\cdot) to talk about the distribution q, not q(m) which is a density. Line 64, just write m ~ q. Similarly for other distributions in the paper. - I don't understand the comment on line 233 - isn't the mean performance ~0.4? Do you mean that one seed got to a return of 1? - I think the paper could be improved too if it is possible to run larger numbers of seeds. The stderr is somewhat meaningless for such small sample sizes like 5.

Reviewer 3



Clarity The paper is clearly written. The main ideas, experiments, and results are described clearly. Originality The identification of noise issues and the proposed solutions appear original. The idea of selective noise injection is new to me, and it addresses the issue with noisy gradients. Although an information bottleneck has been proposed in the past, its evaluation in this setting seems novel. Significance This work should be of use to the deep reinforcement learning community. Techniques such as batch normalization from supervised deep learning do not perform consistently in deep reinforcement learning. This paper's examination of several regularization methods on Coinrun in section 5.3 provides additional evidence of this phenomenon, and the paper provides potentially useful alternatives. Quality The technical aspects of the paper are acceptable. The issues are identified, and the proposed solutions are well motivated. The experiments and evaluation is clear. The presentation could be improved in a few ways as noted below. line 71: Please don't use r_t for the importance sampling ratio in Equation 1 when it also means the reward in equation 5. line 95: Was dropout only applied on the last layer in all the experiments in this paper? Equation 5: The rollout policy should be evaluated at s_t not s_T Equation 12: The number of parameters that are aliased or underspecified is confusing. I see lambda (equation 7), lambda_H, and lambda_V. I also see L^V_{AC} without a definition, but it is perhaps equation 6. However the IBAC algorithms are referenced without a lambda_H or lambda_v term in the experiments Figures 2,3). Please clarify the use of these terms.

[Author Response · NeurIPS 2019]

**Reviewer 1** Thank you for your detailed and in-depth review! *Comparison with other ways of mixing:* In preliminary experiments we tried a schedule in which noise-injection was only introduced after an initial burn-in period. However, the results were much worse, suggesting that it is important for learning good representations that the network is trained on the noisy term from the very beginning. On Coinrun we show that both terms in eq. 7 together substantially outperform using either one of them, supporting the choice we made for SNI. We agree that many other (e.g. continuous) scheduling approaches might also be feasible and we hope to encourage further research in this area. However, we believe we sufficiently show the advantage of our method compared to prior published work by comparing to the existing state of the art for both domains.

*Loose notion of selectivity:* We would like to note that in the mode *with* noise injection, it is done so selectively (i.e. not for $V$ and $\pi_\theta^r$), thus motivating our choice of name. *Motivating eq 7 from section 3:* Issue i) Noise injection can make the model more robust by inducing mistakes that are then improved by gradient updates. However, those mistakes are not useful when acting where they can induce bad actions, so we use the no-noise policy for rollouts $\bar\pi_\theta^r$. Note the faster learning of IBAC-SNI vs. IBAC in Fig. 6 (appendix). ii) Using noise only for updates can lead to a high IS variance, therefore we use the mixture, which empirically improves results (IBAC-SNI with $\lambda = 1$ (no mixture) vs $\lambda = 0.5$). iii) The additional noise through the critic is eliminated in the policy update by using $\bar V_\theta$. We will emphasise these links stronger in the revised paper. *Deep analysis:* We agree and will include plots for variance terms (please see Fig. 1 below for more results on Multiroom, in addition to Fig 3-right already in the paper) and link to videos of trajectories.

*Minor comments:* Those are very helpful, thank you! $\mathcal{Z}$: you're right. Semi-gradient: Yes, we agree this is a better description than "assumed non-changing". $L_{AC}^V$ and $V_\theta(z)$: You're right, we will properly introduce them. $L_{AC}^V$ is eq. 6 and $V(z)$ is the value function taking the latent $z$ as input which depends on $s$ through the encoder. Fig 3: Yes, in the updated figure below. No qualitative change in results. 'Outperforms equal mixing': Yes, it refers to Dropout only.

**Reviewer 2** Thank you for the insightful feedback! We completely agree and will be more explicit about the specific setting we are investigating in the paper, including re-formulating the statements you mentioned. We agree that seeing RL as just minimizing some loss is overly simplistic. We chose this perspective to highlight differences to supervised learning, but will be more nuanced in the updated version of the paper. *Minor points:* Thank you! We will make these changes. Line 233: Yes, successful agents achieve a return of close to 1. We hope the additional plots in Figure 1 show the results better. They also now use 30 instead of 5 seeds on Multiroom and at least 3 seeds on Coinrun.

Figure 1: More seeds and additional plot for Multiroom: In addition to the average return (left), it shows the probability of successfully finding the goal in a layout with a certain number of rooms after training (middle). Note that models are still trained on a mixture of rooms and this separation is only done for testing. The results are averaged over 100 test-layouts per seed and 30 different random seeds per algorithm. The error bars are across seeds. The middle figure shows how much IBAC outperforms alternative regularization techniques especially for more difficult layouts. This shows it allows learning better representations faster and with less available data. Note the *same* model is tested across different room numbers. Especially for more complex layouts (3 rooms) the added stability provided by SNI starts to improve performance. Nevertheless, the main difficulty in multiroom is in learning *general representations*, highlighting the utility of IBAC to do so. SNI becomes necessary on more complex environments like Coinrun (right) where SNI is critical for SOTA performance (note e.g. that multiroom has no 'catastrophic' actions that end the episode). Coinrun results are averaged over 3 seeds (like in [10]) as they are expensive. We will have at least 5 seeds in a future version.

**Reviewer 3** Thank you for your positive and constructive review! Line 71 and Equation 5 have been corrected as you suggested. Line 95: Yes, Dropout was applied only on the last layer in all experiments. We found multiple architectures in the literature but this seems to be the most often used. We will explicitly state this. Equation 12: $\lambda$, $\lambda_H$ and $\lambda_V$ are distinct and we agree that a more distinct notation would be better and will update the paper. The values are given in the appendix but we forgot to mention that $\lambda_H$ and $\lambda_V$ are shared across all experiments and algorithms using the widely used values $\lambda_H = 0.01$ and $\lambda_V = 0.5$. Yes, $L_{AC}^V$ is indeed equation 6. In practice we use PPO, i.e. $L_{PPO}^V$, equation (2). We will clarify the presentation here.

[Meta-Review · NeurIPS 2019]

According to the reviews, this submission is quite easy to evaluate. All reviewers view the paper as presenting a novel and promising technique for regularization via noise injection along with variational information bottleneck. Performance benefits are also shown by state-of-art performance in the CoinRunner domain. Reviewers also found the author feedback quite convincing, as two of the three reviewers raised their overall scores. There were only a few issues mentioned in the revised reviews, and these issues were considered as minor. With final scores of (8, 7, 6) this appears to be an easy accept for NeurIPS.